# Unlocking the Non-deterministic Computing Power with Memory-Elastic Multi-Exit Neural Networks

## ABSTRACT

With the increasing demand for Web of Things (WoT) and edge computing, the efficient utilization of limited computing power on edge devices is becoming a crucial challenge. Traditional neural networks (NNs) as web services rely on deterministic computational resources. However, they may fail to output the results on non-deterministic computing power which could be preempted at any time, degrading the task performance significantly. Multi-exit NNs with multiple branches have been proposed as a solution, but the accuracy of intermediate results may be unsatisfactory. In this paper, we propose **MEEdge**, a system that automatically transforms classic single-exit models into heterogeneous and dynamic multi-exit models which enables **M**emory-**E**lastic inference at the **Edge** with non-deterministic computing power. To build heterogeneous multi-exit models, MEEdge uses efficient convolutions to form a branch zoo and High Priority First (HPF)-based branch placement method for branch growth. To adapt models to dynamically varying computational resources, we employ a novel on-device scheduler for collaboration. Further, to reduce the memory overhead caused by dynamic branches, we propose neuron-level weight sharing and few-shot knowledge distillation(KD) retraining. Our experimental results show that models generated by MEEdge can achieve up to 27.31% better performance than existing multi-exit NNs.

## 1 INTRODUCTION

With the rapid advance of AI and the limited resources at the edge, the efficient use of resources on devices has attracted widespread attention in edge computing [4, 9] and WoT [2, 24, 26, 28]. WoT facilitates the connection and interaction with edge devices over the web. To fulfill the delay requirements, web services on devices usually need to be executed on deterministic computing power (i.e., it cannot be preempted). For example, Concordia [4] assigns dedicated CPU cores to some 5G vRAN processing tasks, in order to obtain deterministic network performance. Because of this, there are bound to be tasks that execute on non-deterministic computing power (e.g., CPU/GPU cores or VRAM which can be preempted). Due to the compute-intensive nature of deep learning inference services and the limited non-deterministic CPU/GPU memory, ensuring the model fits within the available CPU/GPU memory is the key to improving resource utilization in WoT.

However, traditional DNNs cannot use non-deterministic computing power efficiently. Since a traditional DNN only outputs the inference results once at the end, it will not be able to output any result if the computing power is preempted before the inference finishes, wasting the computing power. In order to solve this problem, multi-exit NNs have been proposed in the literature. As the name implies, these multi-exit NNs have multiple exits to output the inference results. This design enables them to output intermediate inference results if the computing power is preempted before the whole inference finishes.

The problem with these multi-exit neural networks is that the accuracy of the intermediate results could be unsatisfactory. In order to improve the inference accuracy, we analyzed the existing multi-exit eural networks and found out that the homogeneous and static structure of the exit models (i.e., branches) was harmful to the inference accuracy. Concretely, static exit models are defined as the models that remain unchanged once they are constructed, while homogeneous exit models mean that all exit branches at different exit locations are identical. As we will show in Section 2, using dynamic and heterogeneous exit models can significantly improve the inference accuracy of multi-exit NNs. Therefore, in this paper, we focus on designing and implementing a system to transform existing single-exit eural networks into multi-exit ones with heterogeneous and dynamic exit models.

There are two main challenges we need to address. First, how to determine the structure, number, and placement of the heterogeneous branches efficiently? There are a large number of possible branch structures, e.g., fully connected (FC) layer or convolution + FC layers, and many possible exit locations of the original NN. It is challenging to automatically select proper branch structures and place them in proper exit locations for various single-exit NNs. Second, how to tackle the high memory consumption problem caused by the dynamic branches? Concretely, using dynamic branches requires the device to store many different versions of branches, resulting in high memory consumption. It is challenging to reduce memory consumption while keeping a sufficient number of branches to achieve high accuracy.

To address the above challenges, we propose **MEEdge**, a system that can automatically transform traditional single-exit models into heterogeneous and dynamic multi-exit models, which enables **M**emory-**E**lastic inference at the **Edge**. To design and place heterogeneous branches efficiently, we utilize efficient convolution techniques to construct alternative branch structures that incorporate FC layers. These superior branches constitute our branch zoo, which we leverage to propose the HPF-based branch placement method. This method facilitates the online construction of a heterogeneous multi-exit model, based on the resources available on edge devices. In order to minimize the memory overhead resulting from dynamic branches, we propose utilizing neuron-level weight sharing. Moreover, we present the few-shot KD technique to increase the inference accuracy. As for the limited and dynamically changing memory resources on devices, we propose to build an on-device scheduler to collaborate with the edge server. Building upon the framework of WoT, this collaboration further extends it by integrating IoT devices with the edge server to distribute computation and storage [24, 26] for for complex web tasks.

Our main contributions can be summarized as follows:

- We propose branch cultivation and HPF-based branch placement method to generate and place proper branches automatically for classical single-exit neural networks.

- We present neuron-level weight sharing and few-shot KD retraining to reduce branch parameters. Further, we design a scheduler for server-device collaboration to minimize the memory overhead of dynamic model inference.
- We implemented our automation system and conducted extensive experiments. The model generated by MEEdge proves to be efficient in multiple cases and can respond quickly to switching in end-to-end performance tests.

The rest of this paper is organized as follows. Section 2 explores the motivation for designing heterogeneous and dynamic multi-exit models. Section 3 provides design details of MEEdge. Section 4 presents the experimental findings. Section 5 discusses related work, and lastly, Section 6 summarizes the work.

## 2 MOTIVATION

### 2.1 Multi-exit vs. single-exit models

Multi-exit networks are widely used in edge computing, which can output inference results before the whole inference finishes. To analyze the relationship between inference time and accuracy of multi-exit networks, we tested the lightweight model (MobileNetV2 [31]), the classic models (LeNet [21], AlexNet [17]), and other works on multi-exit models (BranchyNet [33], MSDNet [8], SPINN [19]) with CIFAR-10 [16] on Raspberry Pi 4B. The execution time and accuracy of different models are shown in Figure 1. Among them, B_AlexNet and S_AlexNet are generated by BranchyNet and SPINN, respectively, and MSDNet7 is produced by MSDNet for seven blocks.

As we can see from Figure 1(a), it is clear that all single-exit inferences (i.e., complete execution of the model) encounter the issue of no inference results before the whole inference process is finished. Besides, we can see that in edge environment, even lightweight models (e.g., MobileNetV2) require a long time to complete the whole inference process. Figure 1(b) shows the multi-exit inference. As we can see from the figure, multi-exit models can output more accurate inference results when they have more inference time.

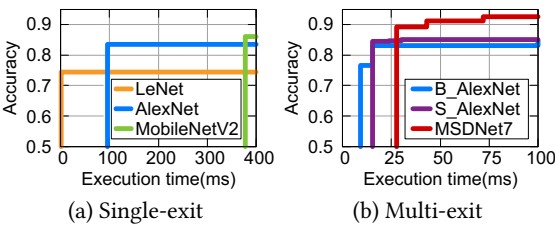

(a) Single-exit  (b) Multi-exit

**Figure 1: Multi-exit inference with early exit can adapt to non-deterministic resources.**

B_AlexNet inserts only two branches at earlier exit points, allowing for results with a shorter execution time. Concretely, B_AlexNet outputs results with an average accuracy of 83.01% in the second branch, using only about 14ms. Note that the original single-exit version of AlexNet (in Figure 1(a)) achieves the final inference accuracy of 83.42%, using 100ms. This example shows how multi-exit models can achieve better inference performance when the computing resource can be preempted at any time. This inspires us to push the convex envelope of the time-accuracy profile of models to the upper left corner by placing proper branch structures.

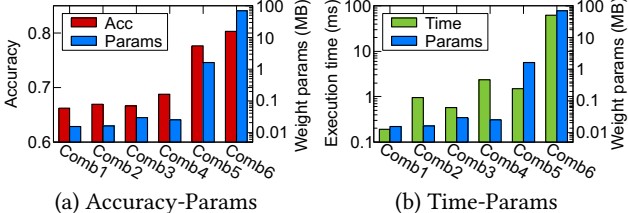

(a) Accuracy-Params  (b) Time-Params

**Figure 2: Profiles of different branching structures including inference accuracy, time, and parameters.**

| | Convolution | Fully connected | Layers |
|---|---|---|---|
| Comb1 | None(1×1) | (in_f, 10) | 0 + 1 |
| Comb2 | DepConv(3×3) | (in_f, 10) | 1 + 1 |
| Comb3 | SpSepConv(7×7) | (in_f, 10) | 2 + 1 |
| Comb4 | DepConv(7×7) | (in_f, 10) | 3 + 1 |
| Comb5 | None(1×1) | (in_f, 2048, 10) | 0 + 2 |
| Comb6 | None(1×1) | (in_f,4096,2048,10) | 0 + 3 |

**Table 1: Details of combinations. Convolutions contain the option of DepConv and SpSepConv [7, 27] with varying kernel sizes. FC layers receive the same input features(i.e., in_f) and differ on hidden layers.**

### 2.2 Static vs. dynamic branches

The placement, number, and structures of branches in multi-exit models are crucial for the inference performance on edge devices. To better analyze the impact of different branches on accuracy and inference time, we illustrate the execution results of LeNet at the second exit for six different branch combinations in Figure 2. Table 1 shows the details, including types of convolutions, configurations of FC layers, and their corresponding number of layers.

Evidently, the unconstrained allocation of memory resources would result in a continual enhancement of branch performance. Yet, memory resources essential for edge computing are both limited and dynamically changing. Dynamically allocating appropriate branches within the available memory ensures model inference instead of being killed. Concretely, a multi-exit model, originally operating within a 30MB memory, loses its inference capability as the available memory diminishes to 20MB.

Besides, we were quite surprised to find out that, enhancing the capacity of convolutions did not result in better inference accuracy, but can have a significant impact on inference time (Comb 1, 2, 3, 4). This is because the CPUs in weak devices are less capable of single-step computation. If the operation exceeds the computing power limit, it must be split into multiple computation steps. Instead of adding convolutional layers, in some cases, it is better to add one FC layer to improve accuracy and reduce execution time (Comb 4, 5). However, using complex FC layers will increase the parameters, resulting in a longer inference time (Comb 6). The proposed convolutions can help mitigate the issue of FC layers by significantly reducing the parameters. However, in our tests, we found that their impact on accuracy was quite limited.

From these experiments, we can see that different branches have a significant impact on the execution time and accuracy of the same model at the same exit location. Therefore, discovering dynamic branches that are both highly accurate and fast within available memory is an important task for multi-exit models.

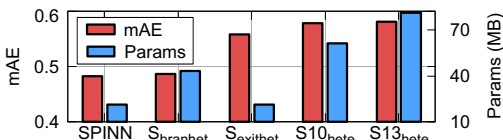

**Figure 3: Heterogeneous branches can significantly improve the inference performance compared to homogeneous ones.**

### 2.3 Homogeneous vs. heterogeneous branches

Previous research with homogeneous multi-exit models [10, 19, 20] overlooks the diversity of branching structures, instead concentrating solely on the placement of branches. We selected VGG-16 [32] as the foundational model and utilized the approach SPINN [19] to construct the multi-exit model. In contrast to homogeneous branches, four heterogeneous branches were designed for comparison. In Figure 3, we show the branch parameters and their corresponding mean Accuracy Expectations (mAE defined in Section 4.1) for the various models.

When the branch structures are heterogeneous ($S_{branhet}$), the performance is improved. And when the exits are heterogeneous ($S_{exithet}$), there is a significant performance improvement while the model parameters are even reduced. For variations that increase the number of heterogeneous branches ($S10_{hete}$, $S13_{hete}$), there is still a performance improvement. Hence, it is beneficial to have heterogeneous branches in the multi-exit model.

## 3 DESIGN

### 3.1 Overview

To tackle the challenges of achieving automatic generations of heterogeneous and dynamic multi-exit models on resource-constraint devices at the edge, we propose MEEdge. Figure 4 shows the overview of MEEdge. For brevity, we utilize the "server" and "device" to refer to edge servers and edge devices without ambiguity. Of course, if devices have ample storage capabilities to store all branches, the online phase can be performed entirely on devices, which is not the focus of this work but allows MEEdge to generalize easily.

To efficiently determine the structure, number, and placement of the heterogeneous branches, the server plays a critical role and requires the involvement of both offline and online phases. (1) MEEdge provides offline training of branch candidates for pre-trained NNs leveraging a predefined *Branch Zoo*. And then it performs *Branch Survival* to eliminate underperforming branches. (2) During the online stage, MEEdge selects and places the heterogeneous branches using a novel HPF-based Branch Placement method, considering the resource budgets sent from devices. Hence, irrespective of the non-deterministic nature of the resources, efficient heterogeneous multi-exit models are assured.

To address the issue of high memory consumption caused by dynamic branches, we propose solutions including both offline operations on servers and online operations on devices. (3) During the online stage, the less capable device will detect dynamic changes in memory and transmit the latency and memory budget to the server, while the server will perform heterogeneous branch selection (i.e. (2)) and on-the-fly branch updating. The device can thus achieve *Memory-Elastic Inference* of dynamic branches. (4) Besides, we propose *Neuron-level Weight Sharing* to reduce the memory overhead

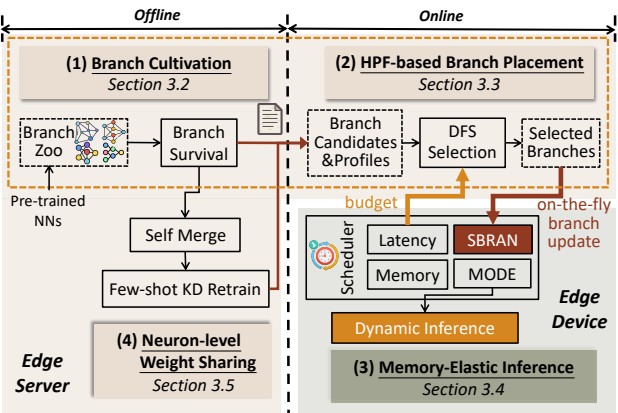

**Figure 4: An overview of MEEdge. Four key techniques involved consist of both offline and online phases, incorporating both servers and devices at the edge.**

of branch parameters at the server during the offline stage. Utilizing branch self-merging and few-shot KD retraining techniques, we can obtain excellent branches. Their profiles will also be considered during the online dynamic branch placement phase.

### 3.2 Branch Cultivation

In this section, we will describe how heterogeneous branch candidates for multi-exit models are generated.

*3.2.1 Branch zoo.* The previous literature adopts branches comprising convolutional and FC layers. However, they are mostly statically specified or homogeneous, which is harmful to the inference accuracy. As we have analyzed, inference performance varies greatly across branches and can not be derived. Thus, we conduct traversals to explore potential branches on different exit points.

We preselected several convolutions and FC structures that could be combined to form effective branch candidates. For convolution kernels, we applied depth-wise, depth-wise separable, spatially separable [7, 27], and dilated convolution [38] instead of 3D-full convolution. They are widely used in the field of Neural Architecture Search (NAS), which has advantages in reducing the parameters and improving the speed and accuracy of inference. Not only that, we get more variants by changing the kernel size (3×3, 5×5, 7×7). For FC layers, we opted for structures characterized by diverse numbers of input features and a flexible range of 1 to 3 layers.

Finally, *Branch Zoo* is formed by different types of convolutions mentioned above, with varying kernel sizes and numbers of layers, as well as different types of FC layers with varying input feature sizes and layer numbers. MEEdge will train diverse branches at each potential exit for the provided NN. During training, crucial characteristics of branches, including weight parameters, inference accuracy, and execution time, are acquired, enabling informed branch placement for constructing heterogeneous multi-exit NNs.

*3.2.2 Branch survival.* For the model that undergoes *Branch Zoo*, numerous distinct trained branches emerge. However, some of them with unqualified inference time and accuracy are not suitable to be candidates. To minimize unnecessary storage and search overhead,

we perform *Branch Survival* to eliminate underperforming branches. Our criterion for underperforming branches is grounded in two considerations: 1) the ability to generate inference results prior to the completion of the original inference; 2) the capacity to yield superior results after the completion of the original inference.

After survival, winning branches will further undergo weight sharing to reduce runtime memory, which will be introduced in Section 3.5. Together with merged excellent branches, they become potential candidates for branch placement in the online stage.

### 3.3 HPF-based Branch Placement

Conducting online branch placement is a crucial component in facilitating server-device collaboration for constructing heterogeneous and dynamic branches. The main challenge lies in effectively selecting and placing high-performance branches from a pool of excellent candidates while adhering to resource constraints. In this section, we will provide the notation and formulation of the selection problem and present the HPF-based branch placement method.

*3.3.1 Notation.* We divide notations into two aspects below.
**Branch Related.** As discussed in Section 3.2.1, there will be lots of branch combinations for each exit. For a model with multiple exits, we consider all original and self-merged branches of all exits as candidates $B$. The profiles of each candidate include structure type (i.e. convolution or FC layer) $B_s$, exit point $B_e$, execution time $T$, inference accuracy $A$, and the number of weight parameters $P$.
**Transmission Related.** During the online stage, data exchange occurs between the server and the device at a transmission rate denoted as $R$. The device detects the expected duration of memory changes and sends the latency budget $L$ and memory budget $M$ to the server. The server then performs searching and sends the selected branches $B_{selected}$ back to the device.

*3.3.2 Problem formulation.* MEEdge focuses on memory-constrained devices. Each incoming task contributes to memory occupancy, which makes it more limited and dynamically changing. To ensure each task adapts to dynamic memory, the optimal heterogeneous branches will be dynamically placed with $L$, $M$, and the assistance of the server. The actual processing time of the server $L_{search}$ requires conversion as shown in Equation 2:

$$L = L_{trans} + L_{search} = \frac{Bits(L+M)}{R} + \frac{Bits(B_{selected})}{R} + L_{search}. \quad (1)$$

The search task, which is constrained by both $M$ and $L_{search}$, seems to be a 2-D knapsack problem. Select certain branches to achieve high performance (i.e., the value of the knapsack) while abiding by memory and latency constraints (i.e., the weight and volume). An intuitive solution would be dynamic programming (DP). However, the wide range of memory and latency values contributes to a heavy and voluminous knapsack, leading to a substantial escalation in time and memory requirements.

It is imperative to establish a metric for the value of branch combinations with $A$, $T$, and $P$. To begin, we focus on plotting the inference time and accuracy in Figure 5. The purple line represents the single-exit inference, and the lower right area (i.e. $acc \cdot time$) can be seen as the accuracy expectation when the given inference time is specified uniformly. Particularly, when the time distribution

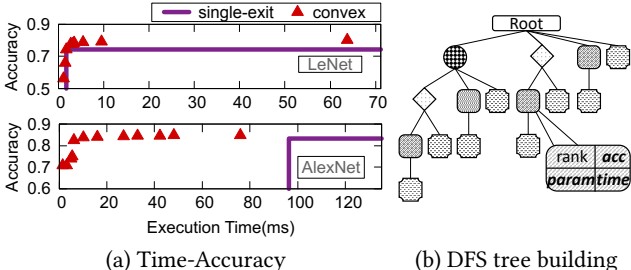

(a) Time-Accuracy      (b) DFS tree building

**Figure 5: (a) Convex branches of multi-exit NNs and single-exit inference. (b) Take four branches as an example.**

---

**Algorithm 1:** Greedy Score Calculation

> **Input:** All candidates branches of a model $B$ and their accuracy $A$, execution time $T$, number of weight parameters $P$.
> **Output:** The sorted branches $B_{sorted}$ and their scores $s$.

1 **begin**
2    **for** $i \leftarrow 0$ **to** $len(B)$ **do**
3      $Per_{best} \leftarrow 0$
4      **for** $j \leftarrow 0$ **to** $len(B)$ **do**
5        **if** $B[j]$ **is not** *appended* **then**
6          $Per' \leftarrow \frac{(A[i]-A_{i-1}) \cdot (T_{i+1}-T[i])}{P[i]}$
7          **if** $Per' > Per_{best}$ **then**
8            $Per_{best}, B' \leftarrow Per', B[j]$
9      $B_{sorted}.append(B')$
10      $s.append(Per_{best})$
11    **return** $B_{sorted}, s$

---

is non-uniform, the calculation should be weighted acc-time integral. Further, we consider $P$ and propose **unit memory accuracy expectation** (i.e. $\frac{acc \cdot time}{params}$) as the metric.

Placing a branch (i.e. red triangle in Figure 5(a)) can achieve a performance improvement that can be simply understood as an increase in the unit memory area. It is evident that placing one branch affects the choice of the other branch, as their areas overlap, which makes searching more challenging.

*3.3.3 Priority-based DFS.* Regardless of search time, the enumeration can always be employed to obtain the optimal branch combination. But as the search space increases significantly, it becomes time-consuming. Heuristic search methods can improve efficiency. However, the interplay among branches exacerbates the difficulty of heuristic searches. In this case, we propose using Depth-First-Search (DFS) to integrate the interplay and the search time.

The core of DFS is the construction of search trees, with each node representing a branch, as illustrated in Figure 5(b). During the tree construction, each node acquires cumulative inference time $l$, memory occupation of parameters $m$, and accuracy expectation $p$ from the root to its position. If a node has a high expectation that satisfies the latency and memory constraints, all nodes on this path together form the branch combination. Properly ordering branches is key to maximizing the efficiency of DFS. We propose to calculate the *priority score* for each branch by Greedy Score Calculation

outlined in Algorithm 1. The algorithm selects the current optimal branches in a greedy manner based on the improvement of unit memory accuracy expectation (i.e. $\frac{\Delta(acc \cdot time)}{params}$) (line 6). Actually, the first path of the tree is the greedy search. Since the tree has multiple branches, using the DFS will yield better results.

Furthermore, there should be a **convex envelop** formed by excellent branches among all branches shown in Figure 5. Theoretically, these branches should be assigned a higher priority score. Hence, we utilize the *convex hull* to fine-tune branches that have already been sorted by priority score before constructing DFS trees, i.e., all sorted convex branches are ranked first keeping the sorted order. Besides, we have tried eight priority score calculation based on Monte Carlo [12] sampling and greedy selection to prove that this calculation we proposed works well in Section 4.3.

The search algorithm is shown in Algorithm 2. The sorted branches are initially obtained by Algorithm 1 (line 2). They are then finetuned by the convex hull (lines 3, 4). The branches are arranged in descending order of priority scores, and then a priority-first multinomial tree is then constructed (line 5). Within the given latency and memory constraints, priority-based DFS (lines 6-19) can always identify the near-optimal combination $P_c$.

The HPF-based Branch Placement enables MEEdge to efficiently update heterogeneous branches of the multi-exit model on devices in a constrained time while adhering to memory constraints.

## 3.4 Memory-Elastic Inference

With the power of the server, weak edge devices can dynamically update the chosen heterogeneous branches based on available memory resources. In this section, we will describe the on-device scheduler we designed and how it can perform memory-elastic inference to adapt to real-time changes in memory.

The scheduler on devices has four components shown in the part (3) of Figure 4, which sends memory and latency budgets to the server for branch placement and receives selected branches.

**Memory Budget.** This module is primarily responsible for the memory changes on the device, regulating how much the memory will change. And the obtained memory budget will be transferred to the server for branch placement.

**Latency Budget.** This module mainly detects the latency budget on the device. For predictable memory changes, if the change occurs after this inference, the latency budget corresponds to the time period between the present moment and the occurrence of the change. To cope with non-deterministic resource changes, i.e. a sudden reduction in available memory occurs during the inference, the multi-exit model should exit early and update branches once changes are detected in order to ensure timely adaptation.

**SBRAN.** This module is used to receive the selected branch combinations and reorganize them with the branches already in memory. For the upcoming memory change, the branches that are not currently stored will be loaded into memory.

**MODE.** This component is utilized to manage two modes of inference related to whether the moment of change can be perceived. One is the *Finite-time Inference*. In this mode, the model stops inference and provides results based on a known latency budget. The other is *Anytime Inference*, which corresponds to non-deterministic resources. The inference will unpredictably exit and give a result.

---

**Algorithm 2:** Priority-based DFS

**Input:** All candidates branches $B$ and their information $A, T, P$ as well as the budget $L_{search}, M$ for the DFS.

**Output:** A better branch combination plan $P_c$.

1 **begin**
2    $B_{sorted}, s \leftarrow \text{Greedy}(B, A, T, P)$    ▷ Get priority scores
3    $B_{convex} \leftarrow \text{convexHull}(B, A, T)$    ▷ Fine tune
4    $B_{sorted} \leftarrow \text{Update}(B_{sorted}, B_{convex})$
5    $Tree \leftarrow \text{Build}(B_{sorted})$    ▷ Building tree
6    **DFS** $(Tree, L_{search}, M)$:
7      $P_c, p' \leftarrow Stack(), 0$
8      $Node \leftarrow Tree.getRoot()$
9      $P_c.push(Node)$
10      **while** $l' < L_{search}$ **do**
11        **for** $node$ in $Node.getChild()$ **do**
12          **if** $node$ **is not** $visited$ **and** $node.p > p'$ **then**
13            $Node, p' \leftarrow node, node.p$
14            $P_c.push(node)$
15        **if** $m' > M$ **then**
16          $Node = P_c.pop()$    ▷ Pruning
17          $p' \leftarrow Node.p$
18    **return** $P_c$

---

With the scheduler and the HPF-base placement, real-time AI tasks on devices can perform online memory-elastic dynamic inference. The dynamic inference here leverages the dynamic inference path of the multi-exit model and dynamic branches. Unlike traditional approaches of offloading or partially offloading models to the server, MEEdge implements branch updating from the server down to the device. It enables smooth model inference on weak devices even when facing sudden changes in memory.

## 3.5 Neuron-level Weight Sharing

Another solution for addressing the issue of high memory consumption is to perform offline operations on the server.

*3.5.1 Branch self-merging.* After *Branch Survival*, there are a large number of branches. Actually, they are composed of similar convolutional and FC layers which can achieve parameter sharing to save storage and memory overhead. Therefore, we propose Neuron-level Weight Sharing after survival. Since the branch structures are much more similar, we adopt MTZ [6] which focuses on neuron-shared layer merging for neural networks with the same number of layers in multi-tasking learning. Taking into account that other existing approaches [13, 18, 22, 29] may overlook the highly similar branch structure, which encounters suboptimal weight-sharing outcomes.

Our primary focus lies in weight sharing across FC layers, which constitute a significant proportion (over 99%) of weight parameters, shown in Figure 6(a). However, merging several branches will inevitably face two FC layers with different numbers of layers or neurons. As shown in Section 4.4.1, merging FC layers with varying numbers of neurons reduces memory occupation but significantly compromises their accuracy. Thus, instead of merging all branches into one hyperbranch, we propose using the neuron-shared *self-merge* method to better leverage the fact that branches themselves

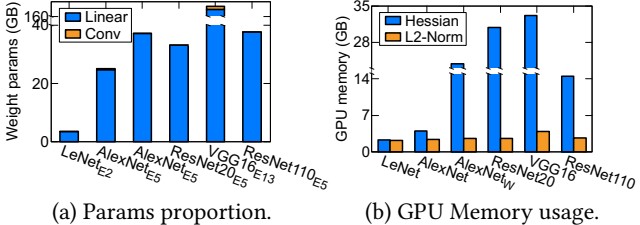

(a) Params proportion.

(b) GPU Memory usage.

**Figure 6: Only FC layers need to be shared and L2-Norm loss calculation can significantly reduce memory usage.**

have the same number of layers and neurons. The *self-merge* can maximize the shared parameters with almost no loss of accuracy, which involves three main steps: 1) Calculate errors between different neurons; 2) Select neuron pairs to be merged; 3) Calculate shared weights for each merged neuron pair.

To measure the difference between different neurons, [6] used the Hessian-based error calculation. However, for the FC layer with a large number of input features, our findings in Figure 6(b) highlight the memory-intensive nature of computations reliant on its Hessian matrix. To allow all branches to be merged efficiently, we propose the L2-Norm loss calculation, and the memory usage is greatly reduced. According to the experimental results in Section 4.4.2, we also find that the merging efficiency with retraining is greatly improved with almost no loss of accuracy.

With these errors, we sorted the individual neuron pairs and merged them in ascending order. As the number of merged pairs increases, the accuracy decreases. Based on our observations, the calculated errors vary considerably for different merging branch pairs, which makes it hard to use a fixed threshold to determine the number of fusion branch pairs. In order to compromise between memory and accuracy, instead of using a fixed threshold, we opt to merge the first third of the neuron pairs and traverse from here to half of them to find the pair with the highest merged accuracy.

To calculate the shared weights of two merged neurons, we simply use the average of two weights, rather than following the Hessian-based approach [6], which proved to be memory-intensive. While the calculation is straightforward, it is highly effective in reducing memory usage and enhancing merging efficiency.

It is inevitable to experience the accuracy loss of self-merged branches although we have tried not to. Therefore, we introduce few-shot KD retraining to recover accuracy.

*3.5.2 Few-shot KD retraining.* Retraining only updates the weights of unshared neurons and the shared weights will remain unchanged. To recover quickly, we select a subset of the origin dataset and use KD to maximize the benefits of these samples.

It is worth emphasizing that KD is not used to distill a new model from the merged model, but self-distillation [39] for efficient recovery. In contrast to [39] considering the knowledge of all exit points, MEEdge only requires the knowledge of the final classifier. We introduce the Kullback-Leibler (KL) divergence between the prediction of the final classifier and the current branch. Moreover, the information derived from the labels is highly significant. Therefore, the Mean Square Error (MSE) between the predicted value and the label also should be incorporated into the loss.

Besides, in order not to interfere with the merged results, the shared weights must be consistent. We add the L1-Norm between the initial and the updated shared weights after back-propagation to the loss. To incorporate KD and shared weight binding while maintaining accuracy, we use the loss function shown in Equation (2) composed of three parts: supervision information from labels, KL divergence for self-distillation, and L1-norm loss of shared weights.

$$\mathcal{L} = \alpha \cdot MSE(y, \hat{y}^i) + (1 - \alpha) \cdot KL(y^c, \hat{y}^i) + \beta \cdot L1(W^s, \overline{W^s}), \quad (2)$$

where $y$ is the label, $\hat{y}^i$ is the prediction of the $i^{th}$ branch, $y^c$ is the prediction of the final classifier, and $W^s$, $\overline{W^s}$ correspond to the shared weights and updated weights by backpropagation, respectively. The $\alpha$ hyperparameter balances ground truth and distilled knowledge, while the $\beta$ hyperparameter determines the accuracy of predictions and stability of shared weights. Based on our experiments, we recommend setting $\alpha$ to 0.5 and $\beta$ to 1.

## 4 EVALUATION

### 4.1 Datasets and setup

**Implementation** We implement MEEdge with *PyTorch*. We use two servers utilizing two NVIDIA GeForce RTX-3090 GPUs and one RTX-3090Ti GPU as the powerful servers for branch training and weight sharing. The server, JETSON XAVIER NX board, is used for online branch selection. Raspberry Pi 4B with 4GB RAM is the device, that communicates with the server through Wi-Fi.

For training details, the learning rates of all multi-exit models and their branch candidates were all set to 0.001. They all trained for 300 epochs and the branches retrained for 200 epochs.

**Datasets** We take into account both image and sensing data modalities. The classic datasets in image classification we use are the CIFAR-10 and CIFAR-100. The sensing dataset we use is Widar3.0.

- The CIFAR-10 and CIFAR-100 datasets [16] contain 32×32 RGB images, composed of 50,000 training and 10,000 testing images, corresponding to 10 and 100 classes, respectively.
- The Widar3.0 dataset [40] is produced by a gesture recognition system utilizing off-the-shelf Wi-Fi devices. It was captured from 5 sensing areas with 5 orientations each. We selected 2,500 samples contributed by 4 users.

**Baselines** We mainly compare MEEdge with the following baselines. The classical NNs to be transformed include LeNet [21], AlexNet [17], VGG-16 [32], ResNet-20, and ResNet-110 [5].

**Single-exit inference** completes the execution of the model.

**Multi-exit inference** is the inference that gets results from all exits, as all branches consist of one FC layer.

**BranchyNet** [33] incorporates a predetermined number of well-designed branches in its assigned location, all through manual intervention. It centers on using LeNet, AlexNet, and ResNet-110.

**SPINN** [19] uses an equal division by FLOPs to determine the placement of six branches [15], and the branch structures are all the same consisting of one convolutional layer and one FC layer.

**Metrics** The accuracy alone fails to measure the performance when dealing with unpredictable inference time due to non-deterministic resources. To simplify the calculation and results demonstration, we proposed the mean Accuracy Expectation (mAE, i.e. $\frac{acc \cdot time}{time_{total}}$) representing accuracy expectation under uniformly distributed time. MEEdge can also accommodate irregular time distributions with

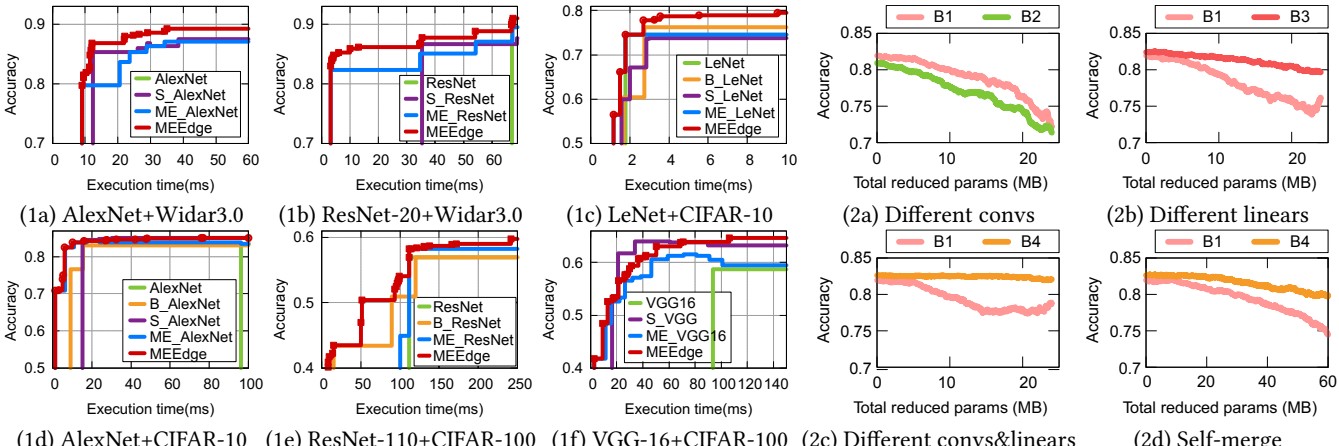

(1a) AlexNet+Widar3.0    (1b) ResNet-20+Widar3.0    (1c) LeNet+CIFAR-10    (2a) Different convs    (2b) Different linears

(1d) AlexNet+CIFAR-10    (1e) ResNet-110+CIFAR-100    (1f) VGG-16+CIFAR-100    (2c) Different convs&linears    (2d) Self-merge

**Figure 7: (1) Finite-time Inference: MEEdge can guarantee better inference performance according to a determined inference time. (2) Self-merge can reduce memory and improve accuracy for all branches compared to twin-merge.**

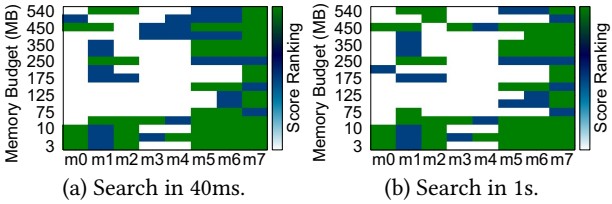

(a) Search in 40ms.          (b) Search in 1s.

**Figure 8: Priority score used by MEEdge (m7) has a superior result in almost all memory budget cases.**

weighted calculation. On the other hand, to evaluate the performance under determined time, we treat the lower right area (i.e. $acc \cdot time$) as the metric and strive to maximize this metric.

### 4.2 Overall performance

*4.2.1 Case study.* **Finite-time Inference:** The models can exit with results according to a determined inference time. We profile the execution time and corresponding accuracy of different models. Figure 7(1) shows the performance of five models on three datasets. The multi-exit models generated by MEEdge always have better results within the same time compared to other baselines. The performance of VGG-16 is suboptimal because the accuracy of branches is constrained by the pre-trained parameters of the single-exit model. **Anytime Inference:** The model needs to output a result immediately at a given arbitrary time during the inference. We use the mAE metric to represent the accuracy expectation MEEdge can achieve for uniformly distributed random inference times. Figure 9(a) shows that MEEdge can improve the mAE among all baselines. And the multi-exit models can indeed provide a solution for non-deterministic resources on devices.

*4.2.2 End-to-end performance.* For the limited and dynamic changing memory, we tested the end-to-end performance of AlexNet and its multi-exit baselines on CIFAR-10 as an example. The transmission rate fluctuates around 85Mbps on average. Figure 9(b) shows performance changes as memory changes 7 times. This change is based on the minimum memory required to ensure the single-exit inference. The values in Figure 9 (b) are the changed budget values.

We can see that AlexNet with MEEdge can flexibly adapt to dynamic changes in memory. For other classic neural networks without display, MEEdge can make them achieve a similar effect facing non-deterministic resources as AlexNet.

### 4.3 Priority-based DFS

We conducted DFS within different memory budgets of eight priority score calculations. A straightforward way is to sort by parameters in ascending order (**m0**). There are also some more thoughtful calculations such as incremental unit memory performance (i.e. $\frac{acc \cdot time}{params}$) (**m1**) and incremental area (i.e. $acc \cdot time$) (**m2**) based on Monte Carlo random sampling, incremental unit memory performance (**m3**) and incremental unit memory performance per memory (i.e. $\frac{acc \cdot time}{params^2}$) (**m4**) based on Monte Carlo importance sampling, incremental unit memory performance (**m6**) and incremental unit memory performance based on greedy search with the convex hull (**m7**). Among them, **m5** is the greedy search.

We plot the results based on all eight priority scores in Figure 8. The **m7** used by MEEdge has a superior result in almost all memory budget cases. It is also worth noting that **m6** is actually an enhancement of the greedy search (**m5**) since DFS will jump to other branches for searching if there is enough searching time.

### 4.4 Neuron-shared branch merging

We selected five branches ( **B1** ($C1F5E0$), **B2** ($C1F4E0$), **B3** ($C13F5E0$), **B4** ($C4F1E0$) and **B5** ($C1F1E1$) ) with different convolution + FC layers of AlexNet on CIFAR-10. The numbers in the descriptions of them correspond to the ID of **C**onv, **F**C, and **E**xit points.

*4.4.1 Self-merge vs. Twin-merge.* Figure 7 (2a)-(2c) shows the twin-merge (i.e. merging of two branches) performance. Compared with twin-merge, the use of self-merge in Figure 7(2d) can reduce memory twice and there is an improvement in the accuracy of each branch. Moreover, it can be inferred from the above that self-merge is much more effective than merging multiple branches.

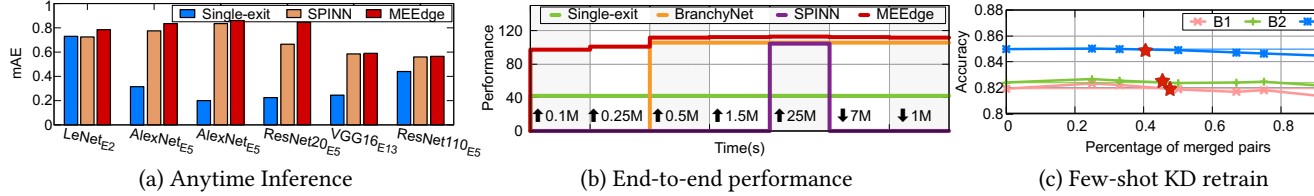

(a) Anytime Inference      (b) End-to-end performance      (c) Few-shot KD retrain

**Figure 9: (a) MEEdge can achieve 7.23-329.79% better performance than single-exit and 0.57-27.31% better than homogeneous and static SPINN facing non-deterministic resources. (b) MEEdge can flexibly adapt to dynamic memory changes. (c) Retraining branches with few merged pairs achieves superior accuracy. And choosing merged pairs from 1/3 to 1/2 traversal is rational.**

*4.4.2 Few-shot KD retrain.* To showcase the performance achieved by few-shot KD retraining, we conducted experiments on the three self-merged branches discussed in Section 4.4.1 using 20% samples of the original datasets. Results in Figure 9(c) indicate that retraining branches with a small number of merged pairs can achieve accuracy exceeding the original accuracy without merging. Furthermore, even when dealing with a large number of merged pairs, there is little loss in accuracy. Meanwhile, the points (red stars) where the accuracy restored by retraining can reach the original accuracy all appear between one-third and half of the merged pairs, which also verifies the rationality of our choice of merged pair numbers.

## 4.5 Overhead

We recorded the branch training and merging overhead on servers, detailed in Table 2. We get 105 branch combinations by selecting 15 convolutions and seven FC layers to form *Branch Zoo*. Few-shot KD retraining was performed after branch self-merging and greatly reduced training overhead. The storage overhead here refers to the sum of the trained and merged branches. The branch profiles only need to be generated once, the overhead is negligible.

| | Training (GPU·days) | Self-merge (GPU·days) | KD retrain (GPU·days) | Storage (GB) |
|---|---|---|---|---|
| $AlexNet_{W3.0}$ | 0.49 | 0.17 | 1.3 | 29.29 |
| $ResNet20_{W3.0}$ | 0.35 | 0.17 | 0.55 | 25.88 |
| $LeNet_{C10}$ | 4.73 | 1.26 | 1.97 | 2.85 |
| $AlexNet_{C10}$ | 6.2 | 4.19 | 4.61 | 20.23 |
| $VGG_{C100}$ | 13.57 | 8.52 | 16.82 | 134.12 |
| $ResNet110_{C100}$ | 11.24 | 3.46 | 8.26 | 30.99 |

**Table 2: Training and merging overhead.**

## 5 RELATED WORK

**Multi-exit Neural Networks.** Multi-exit models have demonstrated the potential for more flexible inference. There are mainly two types. The first is the hand-tuned models. MSDNet [8] builds on top of the DenseNet [11] architecture. And RANet [36] is the extension of MSDNet. The other is to add branches to existing models. BranchyNet [33] was the first to propose adding branches. Later on, HAPI [20], FlexDNN [3], and SPINN [19] automatically designed specific branches on corresponding locations. However, none of these works have taken into account the structural information of the branch itself. This oversight can impact the number and placement of selected branches.

**Cloud-edge Collaboration.** There has been some research on optimizing collaboration between cloud and edge to improve performance and minimize latency. Neurosurgeon [14] was the first to offload part of the model to large servers. Subsequent works [10, 19, 23, 34] inspired by BranchyNet [33] all added early exit mechanisms. Furthermore, [9, 37] even extended the edge AI to extremely weak devices by combining multi-exit NNs. However, pre-divided models may still face issues with non-deterministic resources.

**Neural Architecture Search.** When it comes to designing NNs, the first came to mind was NAS. The initial NNs were manually designed, and then automatic model construction based on RNN-based NAS [30, 41, 42] emerged. Recently, gradient descent methods such as DARTS [25] have become the mainstream method, which has been widely used in the field of deep learning [1]. Currently, NAS is considered mature, but it often requires a significant time to produce high-quality models. Additionally, these models are not scalable enough to accommodate dynamic changes in resources.

**Non-deterministic Resources on Cloud.** The idle resources that go to waste have appeared in cloud computing for a long time. AWS provides two services with different pricing of available resources: resources are not destroyed or preempted at any time. One of the most exciting developments in recent years has been the emergence of serverless computing (i.e., Function-as-a-Service). InfiniCache [35] exploits and orchestrates the memory resources of serverless functions, which achieves 31-96× tenant-side cost savings. However, few works consider non-deterministic resources of AI tasks at the edge we are considering, which has infinite possibilities in the future.

## 6 CONCLUSION

Enabling AI services to adapt to non-deterministic resources among WoT devices, we propose **MEEdge** in this paper. MEEdge is a system that automatically transforms single-exit models into heterogeneous and dynamic multi-exit models for **M**emory-**E**lastic inference at the **Edge**. We address the challenges of efficiently designing and placing heterogeneous branches by proposing Branch Zoo formed by efficient convolutions and the HPF-based Branch Placement. Furthermore, we minimize the memory overhead resulting from dynamic branches by proposing a scheduler on devices to collaborate with the server. For further memory optimization, we propose neuron-level weight sharing and the few-shot KD retraining. Experimental results demonstrate models generated by MEEdge can achieve high accuracy while reducing memory consumption, making it a promising solution for unlocking non-deterministic computing power on resource-constrained devices in WoT.

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
