# OpenReview forum: "Unlocking the Non-deterministic Computing Power with Memory-Elastic Multi-Exit Neural Networks"
_ACM.org/TheWebConf/2024/Conference — TheWebConf24_

### Official Review · Reviewer_1V43 · 2023-11-10

**Novelty:** 5
**Technical Quality:** 5

**Review:**

Paper proposes an approach to create a multi-exit models with a standard single exit networks to handle different inference load on edge devices. It further employs weight sharing an retraining of certain layers with knowledge distillation(KD). The use of KD is not new and has been explored earlier in [1]. The branch zoo concept is not explained clearly, I initially thought it refers to a pretrained model (or layers) but I think this is not the case. Given such a complicated procedure to design a multi-exit model, I am wondering about its feasibility in comparison with a simpler approach like comparing against methods that use entropy of predictions to exit early.

[1] Phuong, Mary, and Christoph H. Lampert. "Distillation-based training for multi-exit architectures." Proceedings of the IEEE/CVF international conference on computer vision. 2019.

**Questions:**

Does High Priority First always try to force the method to exit from earliest layer possible? What if the early exit produce highly uncertain prediction?
How does the performance looks like in comparison with exiting based on entropy?

**Reviewer Confidence:**

3: The reviewer is confident but not certain that the evaluation is correct

**Scope:**

3: The work is somewhat relevant to the Web and to the track, and is of narrow interest to a sub-community

---

### Official Review · Reviewer_hEzM · 2023-11-21

**Novelty:** 5
**Technical Quality:** 5

**Review:**

**Quality.** This paper proposes MEEdge that transforms single-exit models (simple DNNs) into dynamic multi-exit DNNs that are as accurate as single-exit models and enable memory-elastic inference at edge. To achieve its design goals, the paper leverages three novel techniques including dynamic branch construction, neuron-level weight sharing and few-shot knowledge distillation. Therefore, this paper is of good quality in terms of its design objectives and technical contributions.

**Clarity.** This paper is well written and easy to follow. The figures are well formatted and clearly annotated. Some typos exist but the readability is good.

**Significance.** The paper tackles an important limitation of multi-exit models on edge by its novel application of memory overhead minimization technique and knowledge distillation techniques.

In sum, the paper enjoys the following pros and cons:

Pros:
1. The paper proposes a novel multi-exit architecture and applies several techniques to achieve the design goals.
2. The paper is well written.

Cons:
1. Experiments are limited on traditional architectures like ResNet and VGG, the authors should study the scalability to larger and recent vision models (e.g., Transformer-based?).

2. Lack of discussion of potential adaptive threat for this novel multi-exit architecuture. Previous works (e.g., [1-3]) show that existing multi-exit models are vulnerable to adversarial example, backdoor attack and privacy attack. Please discuss whether they can be applied here.
3. Please consider comparison with other model transformations like quantization for edge deployment. Discuss the advantages of multi-exit model.

[1] Hong et al., A Panda? No, It's a Sloth: Slowdown Attacks on Adaptive Multi-Exit Neural Network Inference. ICLR 2021.

[2] Li et al., Auditing membership leakages of multi-exit networks. CCS 2022.

[3] Dong et al., Mind Your Heart: Stealthy Backdoor Attack on Dynamic Deep Neural Network in Edge Computing. INFOCOM 2023.

**Questions:**

See the comments in cons.

**Reviewer Confidence:**

3: The reviewer is confident but not certain that the evaluation is correct

**Scope:**

3: The work is somewhat relevant to the Web and to the track, and is of narrow interest to a sub-community

---

### Official Review · Reviewer_GYiH · 2023-11-22

**Novelty:** 5
**Technical Quality:** 4

**Review:**

This paper proposes memory-elastic multi-exit neural networks. The paper is well written and structured. It is very easy to follow.
Here are the pros and cons of this paper:

Pros:
+ DFS selection in the online stage can work very well for the scenario
+ Multiple existing methods have been compared with the proposed method.
+ The proposed method clearly outperforms the existing methods.

Cons:
- More technical details about Branch zoo and Branch Survival could be added. It is not clear whether NAS is used and how NAS is used. As in Section 5, the disadvantages of NAS have been discussed, but it is not very clear of the technical details of alternative solutions or how NAS is used in the paper to solve those issues. To resolve this, more technical details could be added to Section 3.2.
- In Section 3.2.2, it is declared that "To minimize unnecessary storage and search overhead, we perform Branch Survival to eliminate underperforming branches."As this is offline, the motivation of minimizing unnecessary storage and search overhead could be discussed more, as the search can be done in a server, so need to consider those factors.
- As this paper focuses on the constraints of memory. It would be good to discuss when this could happen, i.e., in which application scenario, there is a memory constraint and in which scenario, the constraints are from other factors, such as computational capability, etc.

**Questions:**

Please see the cons in the review.

**Reviewer Confidence:**

3: The reviewer is confident but not certain that the evaluation is correct

**Scope:**

3: The work is somewhat relevant to the Web and to the track, and is of narrow interest to a sub-community

---

### Official Review · Reviewer_22e2 · 2023-11-23

**Novelty:** 5
**Technical Quality:** 5

**Review:**

The paper addresses the challenge of efficiently utilizing limited computing power on edge devices. It introduces MEEdge, a system that transforms classic single-exit models into heterogeneous and dynamic multi-exit models, enabling Memory-Elastic inference at the edge with non-deterministic computing power​. The authors proposed new approaches such as branch cultivation, HPDF to achieve the goal.

### Pros
* Well-motivated
* I appreciate the illustration used in the paper, they are helpful for readers unfamiliar with this field to navigate such a dense paper
* Automated branch selection and update with dynamic resource constraints

### Cons
* Some errors
    * Line 390 page 4 Eqn 1 is confusing: L appears on both sides,  the line 388 referrs to Eqn 2
    * Line 110 p1 “for for” → “for”

**Questions:**

* What is the overhead (in terms of runtime and storage) of the offline branch cultivation and HPF? Seems like a costly process given the search space
* The back-and-forth communication between device and server can easily take a few hundred ms, which is on par of the single-exit inference time (fig 1). I wonder if the multi-exit approach is truely beneficial if taking into account the server-device communication time?

**Reviewer Confidence:**

2: The reviewer is willing to defend the evaluation, but it is likely that the reviewer did not understand parts of the paper

**Scope:**

4: The work is relevant to the Web and to the track, and is of broad interest to the community

---

### Official Review · Reviewer_pTzV · 2023-11-23

**Novelty:** 5
**Technical Quality:** 5

**Review:**

Summary:
This paper proposes MEEdge, a system that transforms single-exit neural network models into heterogeneous and dynamic multi-exit models for resource-constrained edge devices. The key ideas are: (1) Automatically generate multiple branch candidates with different structures using efficient convolutions. Select high-quality branches through survival analysis to build a branch zoo. (2) Propose a HPF-based branch placement method to select and place heterogeneous branches online according to device resource budgets. This facilitates server-device collaboration for memory-elastic inference. (3) Reduce memory overhead of branches through neuron-level weight sharing and few-shot knowledge distillation retraining on the server. (4) Design an on-device scheduler for collaborating with server to dynamically update branches based on available memory to achieve anytime inference.

Strength:
1. Comprehensively tackles the problem of enabling neural network inference on non-deterministic edge resources through both server-side and device-side techniques.
2. Extensive experiments on image classification and gesture recognition datasets demonstrate the ability to dynamically adapt to resource changes.
3. The proposed techniques such as neuron-level weight sharing, and HPF-based search can be applied more broadly.

Weakness:
1. The end-to-end performance could be analyzed in more complex edge computing scenarios with additional metrics.
2. More analysis could be provided on how the techniques generalize to other model architectures and applications.
3. The security and privacy implications of the server-device collaboration are not discussed.

**Questions:**

please refer to weakness

**Reviewer Confidence:**

3: The reviewer is confident but not certain that the evaluation is correct

**Scope:**

3: The work is somewhat relevant to the Web and to the track, and is of narrow interest to a sub-community

---

### Decision · Program_Chairs · 2024-01-22

**Decision:**

Accept

**Comment:**

This paper proposes a systematic methodology to convert single-exit NNs towards multi-exit branch NNs to suit non-deterministic computing power, with solutions to address heterogenous NN training and non-deterministic computing power scheduling. All reviewers acknowledge the motivations and technical contributions in this paper and provide positive ratings and feedback. The authors are encouraged to incorporate the review comments into the camera-ready version of the manuscript, including the application in more complex scenarios, overhead analysis, etc. I recommend accepting this version to the Web Conference.